# Pre-Operative SARS-CoV-2 Testing in Asymptomatic Heart Transplantation Recipients

**DOI:** 10.3390/biomedicines11082103

**Published:** 2023-07-26

**Authors:** Peter-Paul Zwetsloot, Wouter L. Smit, Niels P. Van der Kaaij, Mariusz K. Szymanski, Manon G. Van der Meer, Linda W. Van Laake, Annet Troelstra, Marjolijn C. A. Wegdam-Blans, Marish I. F. J. Oerlemans

**Affiliations:** 1Department of Cardiology, University Medical Centre Utrecht, 3584 CX Utrecht, The Netherlands; p.p.m.zwetsloot@umcutrecht.nl (P.-P.Z.); m.k.szymanski-2@umcutrecht.nl (M.K.S.); m.g.vandermeer-2@umcutrecht.nl (M.G.V.d.M.);; 2Department of Medical Microbiology, University Medical Centre Utrecht, 3584 CX Utrecht, The Netherlands; w.l.smit-3@umcutrecht.nl (W.L.S.); a.troelstra@umcutrecht.nl (A.T.); m.c.a.blans@umcutrecht.nl (M.C.A.W.-B.); 3Department of Cardiothoracic Surgery, University Medical Centre Utrecht, 3584 CX Utrecht, The Netherlands; n.p.vanderkaaij-2@umcutrecht.nl

**Keywords:** SARS-CoV-2, COVID-19, pre-operative testing, cardiac transplant

## Abstract

Introduction: From the start of the coronavirus disease 2019 (COVID-19) pandemic, international guidelines have recommended pre-operative screening for severe acute respiratory syndrome coronavirus 2 (SARS-CoV-2) before heart transplantation (HTx). Due to the changing prevalence of COVID-19, the chances of false positive results have increased. Because of increased immunity in the population and evolution of SARS-CoV-2 to current Omicron variants, associated mortality and morbidity have decreased. We set out to investigate the yield and side effects of SARS-CoV-2 screening in our center. Methods: We performed a retrospective cohort study in the University Medical Center Utrecht. The study period was from March 2019 to January 2023. All data from patients who underwent HTx were collected, including all pre-operative and post-operative SARS-CoV-2 tests. Furthermore, all clinical SARS-CoV-2 tests for the indication of potential HTx were screened. Results: In the period under study, 51 patients underwent HTx. None of the recipients reported any symptoms of a viral infection. Fifty HTx recipients were screened for SARS-CoV-2. Forty-nine out of fifty patients tested negative. One patient had a false positive result, potentially delaying the HTx procedure. There were no cancelled HTx procedures due to a true positive SARS-CoV-2 test result. Conclusion: Pre-operative SARS-CoV-2 screening in asymptomatic HTx recipients did not lead to any true positive cases. In 2% of the cases, screening resulted in a false positive test result. With the current Omicron variants, in combination with a low-prevalence situation, we propose to abandon pre-operative SARS-CoV-2 screening and initiate a symptom-driven approach for the general viral testing of patients who are called in for a potential HTx.

## 1. Introduction

The coronavirus disease 2019 (COVID-19) pandemic has resulted in increased complexity for advanced heart failure programs and heart transplantation (HTx) care. Transplant program activity decreased in the early stages of the pandemic, including in the Eurotransplant region [1]. At the same time, advanced heart failure patients and HTx recipients faced increased mortality rates, morbidity and complications [2,3,4], which quickly led to an update of the standards of care in the presence of COVID-19. Since the start of the pandemic, guidelines have recommended pre-operative testing for severe acute respiratory syndrome coronavirus 2 (SARS-CoV-2) upon arrival at the hospital when a matched potential donor heart becomes available [5]. This policy was based on the increased morbidity and mortality for SARS-CoV-2-positive symptomatic and asymptomatic patients in the general population undergoing surgery in the early pandemic [6]. Furthermore, the use of immunosuppressive drugs in a relatively frail patient population was also regarded as an increased risk of severe COVID-19 disease after surgery. 

Since the start of the pandemic, the clinical spectrum of COVID-19 has evolved. On a population level, immunity has grown through infections and vaccination [7,8]. In parallel, starting with the Omicron variant (B.1.1.529) in 2021, subsequent subvariants seem to cause less mortality and complications compared to the initial variants of SARS-CoV-2. In many countries, including the Netherlands, obligatory SARS-CoV-2 testing and subsequent quarantining have recently been abandoned for the general population, dealing with SARS-CoV-2 as any other endemic respiratory viral infection with a symptom-driven approach. 

Given the general careful behavior of patients on the waiting list for HTx, the probability of infection with and subsequent diagnosis of SARS-CoV-2 in the absence of (recognized) symptoms is limited. In situations of relatively low COVID-19 prevalence (low pre-test probability), the positive predictive value decreases due to an increase in the number of false positive test results, even with highly specific assays such as the nucleic acid amplification test (NAAT) for SARS-CoV-2 [9]. It has been reported that positive polymerase chain reaction (PCR) tests close to the limit of detection (expressed in high-range CT values) rarely progress to symptomatic COVID-19, although it remains unclear to what extent these results represent a false positive signal or residual SARS-CoV-2 RNA from a previous infection [10]. 

Importantly, borderline positive pre-operative test results have immediate implications for a potential transplant recipient, with potential deferral of the HTx. In this evolving landscape, our current study assesses the added value and false positive rate of pre-operative SARS-CoV-2 screening in potential HTx recipients who are asymptomatic for COVID-19-related symptoms. 

## 2. Materials and Methods

### 2.1. Study Population

We performed a retrospective single-center cohort study at our tertiary advanced heart failure and HTx center, the University Medical Center Utrecht, in the Netherlands. We assessed pre-operative molecular screening of SARS-CoV-2 in all potential HTx recipients from March 2020 to January 2023 in our hospital, who were deemed to be asymptomatic for symptoms related to respiratory infections at the time of screening. All included patients were routinely tested prior to transplantation for the presence of SARS-CoV-2 RNA via nasopharyngeal swabs, using a variety of molecular assays (mono- and multiplex NAAT). The assay of choice depended on the available molecular platform at the time of transplantation and the urgency of testing.

### 2.2. Data Collection

All patients who received their transplant during the selected period were identified. Data were extracted from the electronic health records and included information on baseline characteristics such as age, gender, type of underlying cardiac disease (ischemic versus non-ischemic) and the presence of an LVAD at the time of HTx. Furthermore, data on symptoms of a viral infection (i.e., cough, sneezing, fever) and vaccination status were collected. All SARS-CoV-2 tests performed pre-operatively and post-operatively were used. Any necessary repetitive test results for SARS-CoV-2 (because of weakly positive or borderline positive test results) were also collected. 

To prevent the exclusion of a potential HTx recipient in which the final transplant procedure was aborted (because of quality criteria of the donor heart, logistics or unexpected findings in the potential recipient), we additionally screened all clinically requested SARS-CoV-2 tests in the selected period for the indication “heart transplantation” or any synonym, to potentially identify any patients on the waiting list that were screened pre-operatively but had a deferred HTx for any reason, including a potential positive NAAT for SARS-CoV-2. Furthermore, all advanced heart failure cardiologists were consulted using a questionnaire to assess whether any potential HTx procedure during the investigated period was deferred due to a positive SARS-CoV-2 test result in the absence of symptoms. 

The study was approved by the local medical ethics committee of the University Medical Center Utrecht (non-WMO 18/446) and conducted in accordance with the Declaration of Helsinki. Furthermore, all HTx recipients provided written informed consent for the collection of clinical data as part of a national ongoing quality improvement program.

### 2.3. COVID-19 Screening

The following diagnostic assays were used for the screening of COVID-19: 

Xpert SARS-CoV-2 (Cepheid, Sunnyvale, California, USA An automated multiple real-time reverse transcription polymerase chain reaction (RT-PCR) that is performed on the GeneXpert System, which allows for the quantitative and qualitative detection of SARS-CoV-2 RNA of the N-gene (Nucleocapsid) and E-gene (Envelop). Result and CT-value are calculated and generated via auto-analysis software of the manufacturer (GeneXpert Dx System, Cepheid, Sunnyvale, CA, USA). 

Seegene Allplex™ 2019-nCoV assay (Seegene Inc., Seoul, Korea): A single-tube multiplexed RT-PCR assay for the quantitative and qualitative detection of SARS-CoV-2 RNA, which detects the N-gene, E-gene and RdRp-gene. Result and CT-value are calculated and generated via middleware auto-analysis software of the manufacturer (Seegene Viewer, Seegene Inc., Seoul, Korea). RNA was purified using the Hamilton MicroLAB Star system with the STARMag 96 X 4 Viral DNA/RNA 200 C Kit according to the instructions of the manufacturer (Hamilton Company, Reno, NV, USA). 

ePlex^®^ Respiratory Pathogen Panel 2 (GenMark, Carlsbad, California, USA): A multiplexed nucleic acid test designed to detect a panel of respiratory pathogens, including SARS-CoV-2. It uses hybridizing ferrocene-labeled probes with the sample DNA for amplification. The procedure was carried out according to the instructions of the manufacturer. It is a qualitative test in which no quantitative information (CT-value) is generated. 

ID NOW™ COVID-19 rapid test (Abbott Diagnostics, Lake Forest, IL, USA): An isothermal nucleic acid amplification test that is used in urgent care settings to screen for the presence of SARS-CoV-2 RNA using the RdRp-gene as a monotarget. The procedure was carried out according to the instructions of the manufacturer. It is a qualitative test in which no quantitative information (CT-value) is generated. 

Each assay used was validated in-house in addition to the validations performed by the manufacturer to ensure optimal test characteristics for screening, and showed comparable diagnostic performances. All platforms performed with comparable sensitivity and specificity. 

### 2.4. Statistical Analysis

All data were collected and analyzed in Excel version 16.75.2 (Microsoft, Redmond, WA, USA). Figures were generated using GraphPad Prism version 7 (GraphPad Software Inc., San Diego, CA, USA).

## 3. Results

### 3.1. Patient Characteristics

Of the 51 HTx performed between March 2020 and January 2023, a pre-procedural SARS-CoV-2 PCR result was available in 50 cases included in this study. The mean age of the included population was 51.1 ± 10.5 years, and 36% of the population was female. The etiology of heart failure was ischemic in 16% versus non-ischemic in 84% of the cases. None of the HTx recipients experienced any symptoms associated with COVID-19 before testing. A total of 47 out of 50 patients were vaccinated against SARS-CoV-2, and of which, 30 were pre-HTx and 16 were post-HTx, where 1 patient received his first vaccination pre- HTx and his second vaccination post-HTx. Two patients died after their HTx before vaccinations became available. One patient deliberately declined vaccination (Table 1).

### 3.2. COVID-19 Screening Pre-HTx

Of the 51 included HTx recipients, 1 patient was not tested for unknown reasons, probably related to logistic challenges related to the time of arrival at the hospital and the start of the transplant procedure. In 49 out of 50 patients (98%) tested for SARS-CoV-2, the result was reported as negative. One patient received a borderline positive test result (CT-value ≥ 43, Xpert test) but retested negative upon immediate repeated testing, and was therefore considered to be initially false positive (Figure 1A). The Xpert, Seegene, ePLEX and IDNOW tests were used in, respectively, 38, 4, 6 and 2 cases (Figure 1B). 

In our additional search query for clinically requested SARS-CoV-2 tests, we did not identify any additional tests from patients who had an aborted HTx procedure. Furthermore, the questionnaire among advanced heart failure specialists also did not result in the identification of additional patients in whom the HTx procedure was halted because of a positive SARS-CoV-2 test result. 

### 3.3. In-Hospital Characteristics and Outcomes 

None of the included patients developed symptomatic COVID-19 during their post-operative hospital stay. During the HTx procedure and subsequent post-operative phase in the ICU, no clinical signs of suspected SARS-CoV-2 infection were noted. 

## 4. Discussion

Our current data show that none of the pre-operative SARS-CoV-2 tests in patients turned out to be (true) positive, and they were all deemed to be asymptomatic at their moment of transplantation. A single borderline positive test result turned out to be negative upon repeated testing, potentially delaying the transplantation procedure. This suggests that the added value of pre-procedural SARS-CoV-2 testing in asymptomatic patients admitted to the hospital for a potential heart transplantation procedure is questionable. This is further illustrated by the reduced fatality rates with the new SARS-CoV-2 variants in both the transplant and general population [11,12,13,14], the relatively low prevalence of COVID-19 in the Netherlands and similarities with other (more common) nosocomial viral infections for which routinely pre-operative testing is not recommended in the absence of symptoms [5]. Of note, if future variants of SARS-CoV-2 turn out to be more lethal or contagious despite current measures, this could obviously warrant reinitiation of stricter testing protocols.

During the uncertainties of the early COVID-19 pandemic, HTx programs quickly adapted their protocols to ensure the safe continuation of life-saving HTx for end-stage heart disease. The high prevalence of SARS-CoV-2 combined with high peri- and post-operative risks observed in SARS-CoV-2-infected patients warranted extreme caution and warranted pre-operative testing in all operative procedures at that time, even in the absence of symptoms of a viral infection. Since then, the fatality rate of SARS-CoV-2 and related morbidity has decreased substantially with the introduction of the Omicron variant in the general population [11,12]. Furthermore, vaccination in both the general and HTx population has likely led to decreased morbidity and lethality. 

In recipients of solid organ transplantations, a recent study has shown that the risk of hospitalization (incidence rate ratio of 0.45) and death (incidence rate ratio of 0.17) after SARS-CoV-2 infection decreased significantly with the Omicron variants compared to pre-Omicron variants [13]. Interestingly, a recent study in a cohort of French HTx recipients suggests that the absolute numbers for mortality only slightly decreased during Omicron compared to Delta variants for HTx recipients (9.6% versus 16.1% mortality rates), but not as much as in the general population [14]. The relative mortality compared to the general population for HTx recipients was high in Omicron HTx recipients with an incidence of 289 per 100,000 for HTx (compared to 3.2 per 100,000 in the general population, respectively) [14]. This suggests a worse disease course in HTx recipients compared to the general population with the current Omicron variants [14]. A potential explanation for this phenomenon lies in the relatively low acquired immunity through vaccination in heart transplant recipients due to their immunosuppressive regimens and the suggested immune-evasive properties of Omicron variants [14]. The above studies are relatively contradictory in terms of mortality rates during Omicron variants, and so further studies are needed to clarify these different findings. In light of our current study population and future HTx recipients, patients will predominantly be vaccinated pre-HTx, so will likely have acquired appropriate immunity for current variants, comparable to the general population. However, with new variants and booster regimes, acquired immunity after HTx might still be less than in the general population.

Data on COVID-19 in early post- and peri-operative care after HTx are limited, as current guidelines recommend deferral of the procedure when the SARS-CoV-2 screening test is positive. Several case reports describe cases of transplant recipients who, despite initially testing negative pre-operatively, developed COVID-19 early in the peri- and postoperative courses. A case report from the early pandemic [15] describes two heart transplant recipients who developed COVID-19 early after their transplantation (while being asymptomatic, and testing negative pre-operatively) who ultimately died from COVID-19 due to respiratory failure, despite treatment with antiviral therapy and interruption of the immunosuppressive therapy. A case report from 2022 [16] describes a heart transplant recipient who was diagnosed with COVID-19 due to the Omicron variant 6 days after transplantation with initial mild symptoms, later moderate temporary respiratory deterioration and complete subsequent recovery after treatment, including convalescent plasma. 

The change In approach to COVID-19 in many countries worldwide, including the Netherlands, with the abandonment of quarantining and antigen testing in the general population might influence the situation of our advanced heart failure populations, where among which many are waiting for a potential transplantation. Most likely, the patients that are on the transplant waiting list will keep avoiding contact where possible to reduce the risk of any viral infection in general. The fact that our screening program in two years did not detect any asymptomatic true infections probably underscores this. Systematic pre-operative testing in a ‘very low probability category’ will lead to more frequent false positive results. However, if the case fatality rate rises due to evolution of future variants characterized by higher virulence, or of the population immunity decreases significantly, swift re-implementation of strict test protocols should be reconsidered.

Furthermore, current and pre-COVID-19 protocols already take into account pre-operative screening, including a history of symptoms of (viral) infections, inflammatory markers and a physical examination. Therefore, a potentially harmful viral infection (including SARS-CoV-2) that may affect the post-operative course and outcome after HTx will most probably be detected when patients present themselves for a potential transplant with signs and symptoms of a respiratory infection. The probability of acquiring a SARS-CoV-2 infection during the post-operative period is already minimized due to strict isolation measures that come with post-transplant care. 

## 5. Limitations

There is no registry of aborted HTx procedures in our hospital, and so an aborted procedure based on a positive SARS-CoV-2 test result cannot be fully excluded. For this reason, we performed a questionnaire among all advanced heart transplant cardiologists, who did not report any case of a recalled HTx procedure that was aborted due to a positive SARS-CoV-2 test result. Furthermore, all SARS-CoV-2 test requests in our hospital were screened for HTx indication, which also did not result in any tests associated with an aborted HTx procedure. 

Our study showed a 0% rate of true positives, a 2% rate of false positives and a 0% rate of false negative COVID-19 tests. We are aware that our cohort is small (n = 50), which is a limitation of this study. These low percentages in a relatively small population cannot be extrapolated to make any definite claims on the incidence of SARS-CoV-2 infections in the transplant waiting list population. Large registries are necessary to provide a reliable estimate on the incidence of SARS-CoV-2. 

During the last three years, different PCR platforms have been developed and implemented in our laboratory after thorough evaluation and validation. In the borderline zone of positive results, there was a difference in sensitivity per test. However, real positive results were equally detected independent of the platform used, suggesting that true positive tests should be equally detected across all platforms. 

## 6. Conclusions

Pre-operative SARS-CoV-2 screening in asymptomatic HTx recipients did not lead to any true positive cases, while in 2% of cases, a false positive test resulted in the potential delay of the transplant procedure. Given the risk of false positive results in a low-prevalence situation and the growing body of evidence on lower mortality and morbidity associated with Omicron SARS-CoV-2 variants, a symptom-driven approach for the viral testing of patients who are called in for a potential HTx seems appropriate. In the current situation, we propose to abandon standard pre-operative SARS-CoV-2 testing in asymptomatic HTx recipients. At the same time, we realize the necessity of monitoring the impact of future SARS-CoV-2 variants on vulnerable populations like potential HTx recipients, to adapt screening protocols in a timely manner to dynamic epidemiological circumstances and potentially reinitiate screening in all potential transplant recipients. 

## Figures and Tables

**Figure 1 biomedicines-11-02103-f001:**
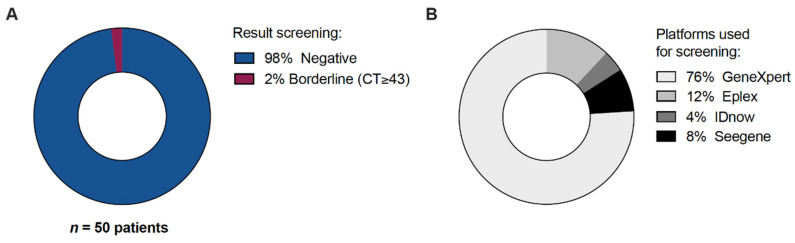
All heart transplant recipients that underwent pre-operative SARS-CoV-2 testing. N = 50. (**A**) Screening result. (**B**) Platforms used for screening.

**Table 1 biomedicines-11-02103-t001:** Baseline characteristics of all included heart transplant recipients *n* = 50.

	Total Patients (*n* = 50)
Mean age (mean ± SD)	51.1 ± 10.5
Gender (male vs. female)	32 vs. 18 (64% vs. 36%)
Type of heart disease (ischemic vs. non-ischemic)	8 vs. 42 (16% vs. 84%)
LVAD in situ pre-HTx	32 out of 50 (64%)
Fever upon presentation	0 out of 50
Other COVID-19 symptoms	0 out of 50
Vaccination status	Vaccinated: 47 out of 50 (94%); 30 pre-, 16 post-, 1 pre/post-HTx Non-vaccinated: 3 out of 50 (6%); 2 died before first vaccination, 1 declined

## Data Availability

Not applicable.

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
