# Peer review of "Pre-Operative SARS-CoV-2 Testing in Asymptomatic Heart Transplantation Recipients"

_biomedicines, 2023, doi:10.3390/biomedicines11082103_

Round 1

Reviewer 1 Report (Previous Reviewer 3)

The improved version of this manuscript can be accepted in the present form.

Author Response

Dear reviewer,

Thank you very much for your response, we are happy to read that our improvements are considered sufficient.

With kind regards,

dr. Oerlemans

Reviewer 2 Report (New Reviewer)

The authors' study performed Pre-operative SARS-CoV-2 screening in asymptomatic HTx recipients, and reported that there was no true positive in the case of asymptomatic patients, but only false positive was confirmed.
If you are asymptomatic, it is recommended that you perform a COVID-19 test if you have symptoms as it is said that COVID-19 screening is meaningless and only false positive.

However, in the case of COVID-19, there are asymptomatic infections and infections in asymptomatic cases, so it can be very dangerous not to conduct a preoperative COVID-19 test at a hospital.

In addition, the number of patients performed in this study is very small, and it should be mentioned that there are other possibilities, that is, the aforementioned asymptomatic COVID-19 infection, and it is possible to spread during the asymptomatic period.

Author Response

Dear reviewer, 

Thank you for your review of our manuscript.

Please find below our point-by-point response to Reviewer 2.

1.

  1. The authors' study performed Pre-operative SARS-CoV-2 screening in asymptomatic HTx recipients, and reported that there was no true positive in the case of asymptomatic patients, but only false positive was confirmed.

Indeed, this is wat we have studied in our cohort of transplant candidates. This is of major importance as well as novel data, as up till now the potential transplant candidates are still being routinely screened in a low endemic situation with prior vaccination for COVID-19 (which is sometimes even mandatory to become listed for heart transplant).

  1. If you are asymptomatic, it is recommended that you perform a COVID-19 test if you have symptoms as it is said that COVID-19 screening is meaningless and only false positive.

Thank you for this comment. We completely agree that with symptoms, a screening for viral and/or respirotory diseases including COVID-19, is indicated. We have emphasized this more in the discussion

(pg 6, lines 239-242);

“Therefore, a potentially harmful viral infection (including SARS-CoV-2) that may affect the post-operative course and outcome after HTx will most probably be detected when patients present themselves for a potential transplant with signs and symptoms of a respiratory infection”

However, in asymptomatic patients this is a matter of debate considering the important consequences of a false-positive result in the current low endemic situation.

As a consequence, if the threat of SARS-CoV-2 increases again (higher incidence or fatality rate), this should obviously be reconsidered as stated in our discussion. We apologize if this was not clear enough, and we added this more clearly to our discussion and conclusion.

(pg 6 lines 234-235 and pg 7 lines 274-275);

“Of note, if future variants of SARS-CoV-2 turn out to be more lethal or contagious despite current measures, this could again warrant stricter testing protocols.”

“…and potentially reinitiate screening in all potential transplant recipients.”

  1. However, in the case of COVID-19, there are asymptomatic infections and infections in asymptomatic cases, so it can be very dangerous not to conduct a preoperative COVID-19 test at a hospital.

Please be referred to the answer above (1B)

  1. In addition, the number of patients performed in this study is very small, and it should be mentioned that there are other possibilities, that is, the aforementioned asymptomatic COVID-19 infection, and it is possible to spread during the asymptomatic period.

Thank you for this comment. We are aware of these limitations. On the other hand, even with this small sample size, one fals-positive patients was identified (and no true-positives), illustrating the complexity vs clincal relevance of this problem in the perspective of a life-saving procedure and potential loss of limited amount of donor organs.

We emphasized this more clearly in the limitation section.

(pg 8, lines 254-255);

“We are aware that our cohort is small (n=50) which is a limitation of this study.”

This manuscript is a resubmission of an earlier submission. The following is a list of the peer review reports and author responses from that submission.

Round 1

Reviewer 1 Report

Authors show their experience on screening testing the presence of SARS CoV2 in HTx.

however as they underlined viral variants as omicron show reduce morbidity andò mortality as several and several articles in Medline; so the article doesn’t offer new insights not offer new insight’s 

Reviewer 2 Report

The work is very interesting and allows you to make new recommendations regarding the process of qualifying heart transplant recipients for surgery in the era of the COVID-19 pandemic.

However, the authors did not clearly formulate the conclusions that follow from such an important observation.

As a reader, I do not understand whether the authors suggest a complete abandonment of preoperative screening for SARS-CoV-2?

or

Do they propose to replace research with quarantine?

Or should only the Antigen test be performed?

I propose to draw a clear conclusion from this important observation.

Reviewer 3 Report

The authors present their own experience in the area of SARS-CoV-2 testing in heart transplant recipients. They show, that in low pre-test probability the risk of false positive results is high. The strategy of universal testing in the current pandemic situation may be even deleterious to the recipients due to the consequences of the false positive result.

The proposed symptoms-driven strategy is reasonable and seems to be cost-effective.

 I have only little concerns.

In the last paragraph of the discussion section the authors discuss OHT from SARS-CoV-2 positive donors, but the topic of the study is SARS-CoV-2 testing of the recipients. There is slight or no connection with the current study.